# FastGHA: Generalized Few-Shot 3D Gaussian Head Avatars with Real-Time Animation

**Xinya Ji**[1,2]**, Sebastian Weiss**[3]**, Manuel Kansy**[3]**, Jacek Naruniec**[3]**, Xun Cao**[1]**,
Barbara Solenthaler**[2]**, Derek Bradley**[3]
[1]Nanjing University, [2]ETH Zürich, [3]DisneyResearch|Studios

## Abstract

Despite recent progress in 3D Gaussian-based head avatar modeling, efficiently generating high fidelity avatars remains a challenge. Current methods typically rely on extensive multi-view capture setups or monocular videos with per-identity optimization during inference, limiting their scalability and ease of use on unseen subjects. To overcome these efficiency drawbacks, we propose FastGHA, a feed-forward method to generate high-quality Gaussian head avatars from only a few input images while supporting real-time animation. Our approach directly learns a per-pixel Gaussian representation from the input images, and aggregates multi-view information using a transformer-based encoder that fuses image features from both DINOv3 and Stable Diffusion VAE. For real-time animation, we extend the explicit Gaussian representations with per-Gaussian features and introduce a lightweight MLP-based dynamic network to predict 3D Gaussian deformations from expression codes. Furthermore, to enhance geometric smoothness of the 3D head, we employ point maps from a pre-trained large reconstruction model as geometry supervision. Experiments show that our approach significantly outperforms existing methods in both rendering quality and inference efficiency, while supporting real-time dynamic avatar animation.

## 1 Introduction

Over the last few years the field of digital avatars has seen tremendous growth, fueled primarily by the advent of neural rendering (Mildenhall et al., 2021; Lombardi et al., 2019) and 3D Gaussian Splats (3DGS) (Kerbl et al., 2023) in particular. The simple parameterization, tractable reconstruction, and real-time novel view rendering of 3DGS provides a suitable representation for photoreal digital human avatars. For this reason, Gaussian-based avatars have been extensively explored in several recent works (Kirschstein et al., 2025; Gong et al., 2025; He et al., 2025; Chu & Harada, 2024).

Initial digital avatar methods require synchronized multi-view video or long monocular data for the subject to be reconstructed (Hong et al., 2022; Qian et al., 2024; Giebenhain et al., 2024; Wang et al., 2025a; Zielonka et al., 2025; Ma et al., 2024). While this approach proved the ability to represent digital avatars using Gaussian-based representations, this avenue is clearly not scalable. Some works reduce the data requirements by synthesizing novel views first and then optimizing a Gaussian avatar on these synthetic observations (Taubner et al., 2025; Zhou et al., 2025; Tang et al., 2025; Yin et al., 2025; Gu et al., 2025). While this alleviates the need for dense captures, it still requires long per-identity reconstruction. Another line of research involves training generative models with 3D head priors, but still requires fitting an identity code for each new person at inference (Zheng et al., 2025; Sun et al., 2023; Tang et al., 2023; Xu et al., 2025). The most efficient and attractive approach that has emerged so far is to build the avatar in a feed-forward way given only a small set of images of the subject. While this requires pre-training a generalized model on a large corpus of human face data, it enables instantaneous reconstruction and novel-view rendering of unseen subjects at inference time. Inspired from the success of large reconstruction models (Xu et al., 2024a), recent work such as Avat3r (Kirschstein et al., 2025) and Facelift (Lyu et al., 2024) leverage a feed-forward network to fuse sparse input views into 3D Gaussian head representations. However, these methods either lack support for controllable facial animation or suffer from slow animation speed and limited reconstruction fidelity.

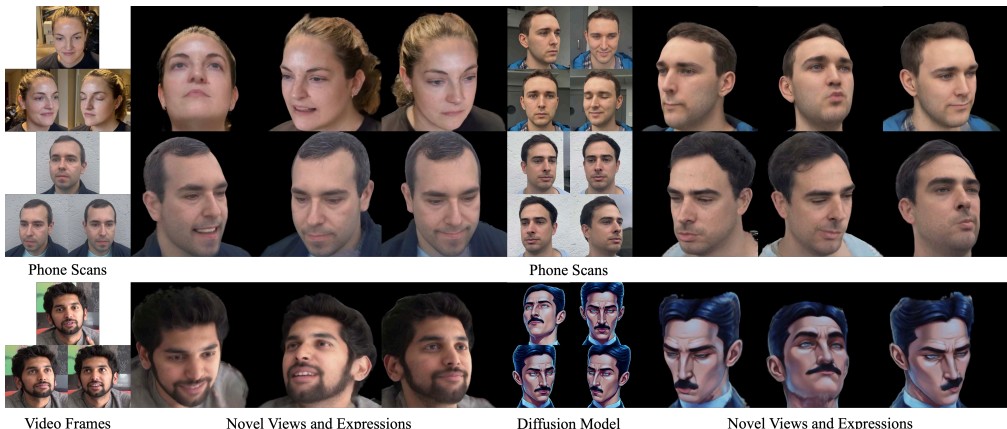

Figure 1: We present FastGHA, a feed-forward method that generates Gaussian head avatars from few-shot input images and animates them in real time.

In this work, we propose *FastGHA*, a new architecture for generalized few-shot 3D Gaussian head avatar reconstruction that allows real-time animation. Our network is a feed-forward approach that takes only a few images of a subject from well-distributed views and arbitrary expressions and learns a high quality animatable avatar. We infer a per-pixel Gaussian representation, enabling the preservation of fine-grained head details. To achieve high-fidelity and efficient animation, we adopt a two-stage pipeline to first reconstruct a canonical Gaussian head from the input images, followed by a lightweight deformation network to animate the Gaussians based on descriptive expression codes. In this way, our model learns to disentangle the different facial expressions in the input images by effectively mapping the images to a neutral canonical Gaussian head before applying animation. For high-quality reconstruction, we leverage features extracted from a pre-trained diffusion VAE (Rombach et al., 2022) and DINOv3 (Siméoni et al., 2025), and introduce a transformer-based feed-forward module to fuse multi-view information from these features. For real-time animation, we extend the explicit Gaussian representation with per-Gaussian features and introduce a lightweight MLP-based dynamic network to predict fine-scale Gaussian deformations from expression codes. Furthermore, to enhance geometric smoothness of the 3D head and the robustness of our model, we employ point maps predicted from a pre-trained transformer (Wang et al., 2025b) as geometry supervision. We train the network on large scale multi-view head video datasets in an end-to-end way. Our method allows real-time animation of Gaussian head avatars created in a feed-forward manner, with reconstruction in less than one second. Extensive experiments also show that *FastGHA* outperforms Avat3r and other state-of-the-art approaches in both reconstruction fidelity and animation efficiency.

Specifically, we offer the following contributions:

- We propose *FastGHA*, a framework for generalizable 3D Gaussian head avatar reconstruction from few-shot images with measurable quality improvement over current state-of-the-art methods;
- We achieve real-time animation by introducing a lightweight MLP conditioned on extended per-Gaussian features to learn fast pixel-wise 3D Gaussian deformations;
- We formulate a geometry prior from VGGT as a regularization loss during training to improve the 3D head consistency and robustness.

## 2 RELATED WORK

**Animatable 3D Head Avatars.** Generating animatable 3D head avatars from a few input images is a challenging task that typically requires accurate head geometry reconstruction and a robust method for capturing dynamic expressions. Early works (Khakhulin et al., 2022; Xu et al., 2020) rely on parametric face models such as 3DMM to reconstruct and animate human heads. These methods are limited by rendering quality and fail to produce realistic animation. With the advent of Neural

Radiance Fields (NeRFs) and subequent 3D Gaussian Splatting (3DGS), high-quality avatar reconstruction with real-time rendering became feasible. Most approaches (Gafni et al., 2021; Gao et al., 2022; Hong et al., 2022; Ki et al., 2024; Ma et al., 2024; Xu et al., 2024b; Qian et al., 2024; Giebenhain et al., 2024; Dhamo et al., 2024; Tretschk et al., 2021; Wang et al., 2025a; Zielonka et al., 2025) require expensive training on a long monocular or multi-view video sequences for each subject, making them impractical in many scenarios. Some methods (Taubner et al., 2025; Zhou et al., 2025; Tang et al., 2025; Yin et al., 2025; Gu et al., 2025) bypass the extensive data requirement, by first generating novel-views from a limited set of images, and then optimizing a Gaussian avatar using these synthetic observations. This approach, however, does not solve the long optimization time for new characters. Another line of research (Zheng et al., 2025; Sun et al., 2023; Tang et al., 2023; Xu et al., 2025) leverages large-scale video datasets to learn 3D head priors by training generative avatar models. During inference time, an identity code is fitted to a new, unseen person. This process is time-consuming, thus restricting the set of possible application scenarios. Recently, several works (Kirschstein et al., 2025; Gong et al., 2025; He et al., 2025; Chu & Harada, 2024; Zhao et al., 2024; Chu et al., 2024) explored feedforward avatar reconstruction in a few- or single-shot setting. Most single-shot methods trained on large monocular video datasets (Gong et al., 2025; He et al., 2025; Gafni et al., 2021) lack 3D consistency and focus primarily on the frontal view. Most recently, Avat3r (Kirschstein et al., 2025) introduced an animatable, feedforward Gaussian reconstruction model based on four arbitrary views. However, the inaccurate geometry predicted by a general reconstruction model is directly injected into the network through skip-connections, which negatively affects the reconstruction fidelity. Moreover, they use cross attention blocks to model facial animation from expression codes, resulting in more computation and slow rendering speed. Specifically, it is not possible to achieve real-time novel animation with their architecture. In this paper, we employ a similar feedforward paradigm as Avat3r (Kirschstein et al., 2025) and support an arbitrary number of input views. We use a large reconstruction model, VGGT (Wang et al., 2025b), as a geometry prior during training, avoiding the error propagation during inference. We also design a lightweight MLP to model Gaussian deformations from expression codes, enabling real-time animation with high fidelity.

**2D Facial Animation.** In recent years, the development of deep generative models has led to rapid advancements of photoreal 2D facial animation methods. Early work (Qiao et al., 2018; Siarohin et al., 2019; Wang et al., 2021; Tewari et al., 2020) take GANs (Goodfellow et al., 2020; Karras et al., 2020; 2019) as the backbone for motion generation. Though computationally efficient, these methods lack diversity and suffer from poor generalization to occluded areas and restricted expressiveness for intricate facial dynamics. Recently, diffusion-based generative models (Song et al., 2021a;b; Wang et al., 2024a) have demonstrated remarkable capabilities and superior quality in many image generation tasks. Large pre-trained models, such as Stable Diffusion (Rombach et al., 2022), have spurred numerous applications leveraging their robust model priors. Early works (Guo et al., 2024; Hu, 2024; Xu et al., 2024c; Guan et al., 2024) explored the animation task by extending the pre-trained model from image generation to video generation by introducing temporal components. DreamPose (Karras et al., 2023) introduces a dual clip-image encoder to animate a person given a set of input poses. Other methods (Guan et al., 2024; Xu et al., 2024c; Hu, 2024) resort to a ReferenceNet to generate full body animation based on the rough joint estimates. Video diffusion models (Ho et al., 2022) now emerged as a backbone for high-quality facial animation, with explicit modeling of temporal correlations to produce smooth and photorealistic videos from prompts or given driving frames. However, these methods (Cui et al., 2025; Blattmann et al., 2023; Yu et al., 2025) come with high computational cost, limiting their efficiency in real-time scenarios. Moreover, since they operate in the 2D domain without explicit 3D constraints, these methods struggle with geometric artifacts under large view changes. In contrast, our approach reconstructs an explicit 3D Gaussian avatar, which enables controllable animation under arbitrary viewpoints and expressions in real time.

## 3 FASTGHA METHOD

As illustrated in Figure 2, the goal of our model is to generate a high-fidelity and animatable head avatar from few-shot input views of a target subject using dynamic 3D Gaussians controlled by expression codes (Section 3.1). We first reconstruct a canonical Gaussian point cloud from the input images and their camera parameters. Here, "canonical" means an un-posed Gaussian head with little to no facial expression, although we do not supervise this intermediate output. Note

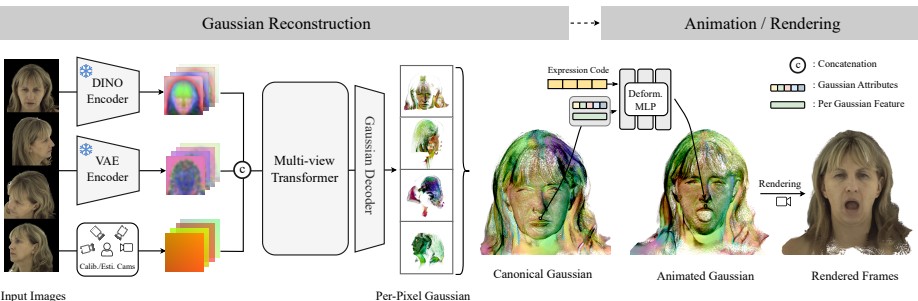

Figure 2: Overview of our method. Given a few input images with arbitrary views and expressions, we first extract multi-view features with pre-trained models and then train a multi-view transformer network that projects these features into 3D to reconstruct a canonical Gaussian head avatar. To enable real-time animation, we introduce a lightweight MLP that deforms the Gaussians according to the expression code.

that the input images can portray any camera view or facial expression, as long as the views are well-distributed, without requiring a controlled multi-view capture system. Then, we introduce a lightweight multi-layer perceptron (MLP), which deforms the canonical Gaussians according to FLAME expression codes (Li et al., 2017) to handle the facial dynamics in real time. To achieve this feed-forward prediction, we train our model on a large multi-view and multi-expression dataset using a combination of photometric loss and a geometric loss that employs a large foundation model VGGT (Wang et al., 2025b), see Section 3.2.

## 3.1 ANIMATABLE GAUSSIAN HEAD AVATARS

**3D Gaussian Representation.** In 3D Gaussian Splatting (3DGS) (Kerbl et al., 2023), the scene is represented by a set of Gaussian ellipsoids $\mathcal{G}$, each defined by its position $\mathbf{X}$, multi-channel color $\mathbf{C}$, rotation $\mathbf{Q}$, scale $\mathbf{S}$ and opacity $\alpha$. Given camera parameters $\mu$, these Gaussians can be rasterized and rendered (collectively denoted as $\mathcal{R}$) into novel-view images $I$. This process can be described as:

$$\mathcal{G} = \{\mathbf{X}, \mathbf{C}, \mathbf{Q}, \mathbf{S}, \alpha\}, \qquad I = \mathcal{R}(\mathcal{G}, \mu). \tag{1}$$

In order to faciliate the learning of Gaussian dynamics driven by expression codes, we augment the default Gaussian representation with a per-Gaussian feature $\mathbf{f} \in \mathbb{R}^{N \times 32}$:

$$\mathcal{G}_f = \{\mathbf{X}, \mathbf{C}, \mathbf{Q}, \mathbf{S}, \alpha\} \cup \{\mathbf{f}\}. \tag{2}$$

These additional features encode high-level semantic information that guide the update of Gaussian attributes during animation and are used below to deform the Gaussians for novel expressions.

**Feature Extraction and Reconstruction.** Different from person-specific optimization-based approaches (Xu et al., 2024b; Giebenhain et al., 2024; Dhamo et al., 2024), our model aims for feed-forward reconstruction, enabling fast head avatar reconstruction. Thus, instead of using template Gaussians rigged to a 3DMM (Li et al., 2017; Gerig et al., 2018) face template, we draw inspiration from the recent success of large reconstruction models (LRM) (Xu et al., 2024a; Zhang et al., 2024b; Tang et al., 2024; Kirschstein et al., 2025) and directly predict Gaussian primitives for each pixel of the input images, see Figure 2–Reconstruction. Following the paradigm of a LRM, we first extract per-view features using pre-trained models, then apply a Vision Transformer (ViT) to capture multi-view correspondence and a decoder to regress Gaussian attributes.

Specifically, we adopt the VAE from SD-Turbo (Sauer et al., 2024) for color features and DINOv3 (Siméoni et al., 2025) for semantic features to encode RGB images $I_{in} \in \mathbb{R}^{V \times 3 \times H \times W}$ into low-dimensional latent features, where $V$ represents the number of input images. Camera poses are represented as Plücker ray maps following Zhang et al. (2024a;b), which takes the form of image-like 2D data. The pre-trained encoder weights are frozen during training to ensure stable and semantically meaningful features. For DINOv3, we apply extra convolutional layers to fuse the

features derived from different layers and obtain a single feature map $F_{dino}$ that matches the spatial dimension of the VAE feature map $F_{enc}$. We refer to the supplementary material for details on the different feature layers. Then, we concatenate the image features and the Plücker ray coordinates map $P_{cam}$:

$$\mathcal{V} = [F_{dino}; F_{enc}; P_{cam}] \in \mathbb{R}^{V \times (C_d + C_e + 6) \times H_f \times W_f}, \tag{3}$$

where $[\cdot; \cdot]$ denotes channel-wise concatenation, $C_d, C_e$ denote the channel dimensions of the DINOv3 and VAE feature maps respectively, and $H_f = H/8, W_f = W/8$. The concatenated latent $\mathcal{V}$ is fed into a Multi-view Transformer consisting of multiple self-attention layers to integrate cross-view cues and model geometric correspondences. Here, we follow the implementation of GRM (Xu et al., 2024a), where each self-attention layer operates on $V \times H_f \times W_f$ tokens and outputs a feature vector with the same length. The resulting latent tokens $\tau$ are then decoded via a modified SD-Turbo VAE decoder, where the input and output channels are expanded to accommodate $\tau$ and regress the Gaussian attributes while keeping the pre-trained inner layers as initialization. During training, we fine-tune the entire decoder. The decoder outputs a set of Gaussian image maps that contain per-pixel 3D Gaussian parameters. We regress the map for each input view, where the 3D Gaussians represent the head in a learned canonical expression. By design, the canonical head is devoid of expression and thus does not correspond to any of the input expressions, allowing animation to happen downstream. These maps are fused into a single canonical 3D Gaussian head, $\mathcal{G}_f^c$.

**Gaussian Head Animation.** To achieve real-time animation of pixel-wise Gaussians, we leverage a Multi-layer-Perceptron (MLP) $\mathcal{D}$ to model the facial dynamics based on FLAME expression parameters. Specifically, $\mathcal{D}$ takes a FLAME expression code $\mathbf{z}_{exp}$ of the desired target expression along with the canonical Gaussian head $\mathcal{G}_f^c$ including learned per-Gaussian features and predicts offsets $\delta_{\mathbf{z}}$ for the Gaussian attributes:

$$\delta_{\mathbf{z}} = \mathcal{D}(\mathcal{G}_f^c, \mathbf{z}_{exp}). \tag{4}$$

Specifically, we predict expression-specific offsets to the position $\mathbf{X}$ and color $\mathbf{C}$ of the Gaussians. Note that $\mathcal{D}$ acts independently on each Gaussian point, enabling efficient and parallelizable deformation. The deformed Gaussians can then be rendered to get the final image $I_{out}$, alpha mask $M_{out}$ and depth map $D_{out}$ (used for supervision) at arbitrary camera views through the differentiable rasterizer $\mathcal{R}$.

## 3.2 LOSS FUNCTION

We supervise the training with photometric losses on a set of novel view images (see Section 4.1 for details). Our losses include an RGB loss and a SSIM loss:

$$\mathcal{L}_{RGB} = ||I_{out} - I_{gt}||_1, \quad \mathcal{L}_{SSIM} = SSIM(I_{out}, I_{gt}). \tag{5}$$

We also use a perceptual loss (Zhang et al., 2018b; Johnson et al., 2016) as well as a silhouette loss to encourage the 3D shape consistency:

$$\mathcal{L}_{perc} = VGG(I_{out}, I_{gt}), \quad \mathcal{L}_{sil} = ||M_{out} - M_{gt}||_1. \tag{6}$$

Recent work (Kirschstein et al., 2025; Shi et al., 2025; Jiang et al., 2025) has shown the benefit of including a geometry prior predicted by a 3D geometry model (Wang et al., 2025b; 2024b) for Gaussian-based reconstruction. Therefore, we introduce a geometric loss derived from a pre-trained VGGT (Wang et al., 2025b) backbone. Different from Avat3r (Kirschstein et al., 2025), which directly takes the predicted point map as input, we incorporate the geometry prior as a regularization loss during training. This approach alleviates the requirement of the network to explicitly compensate for discontinuities and artifacts in the prior. To ensure the consistency between the geometry prior and our Gaussian avatar, we first align the VGGT point cloud with the ground truth 3D face mesh (more details are presented in the supplementary document). Then, the aligned points are reprojected by camera $\mu$ to infer a depth map, which serves as the ground-truth $D_{gt}$. The geometric loss is formulated as:

$$\mathcal{L}_{geo} = ||D_{out} - D_{gt}||_1. \tag{7}$$

Table 1: Quantitative comparison with state-of-the-art methods on Ava-256 and Nersemble datasets.

| Method | Ava-256 | | | | | NeRSemble | | | | |
|---|---|---|---|---|---|---|---|---|---|---|
| | PSNR ↑ | SSIM ↑ | LPIPS ↓ | CSIM ↑ | AKD ↓ | PSNR ↑ | SSIM ↑ | LPIPS ↓ | CSIM ↑ | AKD ↓ |
| InvertAvatar | 14.2 | 0.36 | 0.55 | 0.29 | 15.8 | 15.1 | 0.47 | 0.46 | 0.45 | 8.2 |
| GPAvatar | 19.1 | 0.70 | 0.32 | 0.26 | 6.9 | 20.8 | 0.76 | 0.30 | 0.54 | 6.3 |
| Avat3r | 20.7 | 0.71 | 0.33 | 0.59 | **4.8** | - | - | - | - | - |
| Ours (Ava-256) | 22.2 | 0.76 | 0.24 | 0.71 | **4.8** | 22.3 | 0.78 | 0.27 | 0.70 | 4.8 |
| Ours (both) | **22.5** | **0.77** | **0.23** | **0.73** | **4.8** | **24.0** | **0.81** | **0.24** | **0.77** | **4.4** |

Overall, the total loss function is formulated as:

$$\mathcal{L} = \mathcal{L}_{RGB} + \lambda_{SSIM}\mathcal{L}_{SSIM} + \lambda_{perc}\mathcal{L}_{perc} + \lambda_{sil}\mathcal{L}_{sil} + \lambda_{geo}\mathcal{L}_{geo}, \tag{8}$$

where $\lambda$ denoting the weights of each term, and are set as follows: $\lambda_{SSIM} = 1, \lambda_{sil} = 1, \lambda_{perc} = 0.5$ and $\lambda_{geo} = 0.5$.

## 4 EXPERIMENTS

In the following, we start by describing our training data, implementation details and comparison baselines in Section 4.1. We then demonstrate our results, comparisons and analysis in Section 4.2. Ablation studies that justify our main design decisions are given in Section 4.3, and we demonstrate applications of our method in Section 4.4.

### 4.1 EXPERIMENTAL SETUPS

**Dataset.** Our model is trained on a combination of two multi-view video datasets, Ava-256 (Martinez et al., 2024) and Nersemble (Kirschstein et al., 2023). The Ava-256 dataset contains 256 subjects captured with 80 cameras, from which we use only 40 cameras with frontal viewpoints for training. The Nersemble dataset has a total of 425 subjects with 16 cameras. For each video sequence, we first crop the image to $512 \times 512$ resolution, remove the background (Lin et al., 2021) and then employ off-the-shelf head tracking tools (Zielonka et al., 2022; Qian et al., 2024) to obtain FLAME expression weights Li et al. (2017) with jaw and eye poses as the per-frame expression codes $\mathbf{z}_{exp}$. Similar to Kirschstein et al. (2025), in each training iteration, we randomly sample four images of the same subject with different expressions and camera poses as input to the network. We additionally sample eight supervision images with the same expression, which is potentially different from the four input images. From these eight supervision images, four match the input camera views and four are from novel views, to provide multi-view and novel-expression supervision.

**Implementation details.** We train the network using the ADAM optimizer (Kingma & Ba, 2014) with a learning rate of $5e^{-5}$, except for the MLP $\mathcal{D}$, where we use a higher learning rate of $5e^{-4}$. The training takes roughly four days for 400k steps on four H800 GPUs with a batch size of 1 on each GPU. During inference, the first reconstruction part of our network only needs to be performed once for a given set of input images and takes less than one second. After that, our method achieves real-time animation and rendering on a single H800 GPU.

**Baselines.** We compare our approach with other state-of-the-art methods including InvertAvatar (Zhao et al., 2024), GPAvatar (Chu et al., 2024) and Avat3r (Kirschstein et al., 2025), which all reconstruct 3D head avatars from few-shot input images in a single forward pass. InvertAvatar uses 3D GAN inversion for head avatar reconstruction. GPAvatar is based on the tri-plane representation (Chan et al., 2022) and uses a FLAME point cloud as prior to sample 3D points for animation. Avat3r is the most related to our work, as it directly regresses pixel-wise 3D Gaussians from the images. However, Avat3r incorporates the expression parameters directly into the cross-attention model for reconstruction, which comes at the cost of a slow animation speed of only 8 fps.

**Metrics.** To measure the image quality, we employ Peak Signal-to-Noise Ratio (PSNR), Structural Similarity Index (SSIM) (Wang et al., 2004) and Learned Perceptual Image Patch Similarity (LPIPS) (Zhang et al., 2018a) between the synthesized images and the ground truth images. We then use ArcFace (Deng et al., 2019) to compute the cosine similarity of the identity features for identity

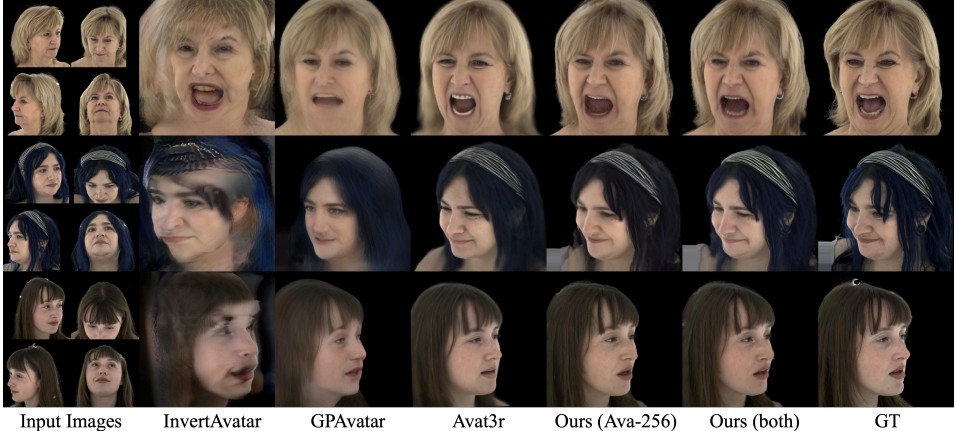

Input Images    InvertAvatar    GPAvatar    Avat3r    Ours (Ava-256)    Ours (both)    GT

Figure 3: Qualitative reconstruction comparison on Ava-256 held-out subjects.

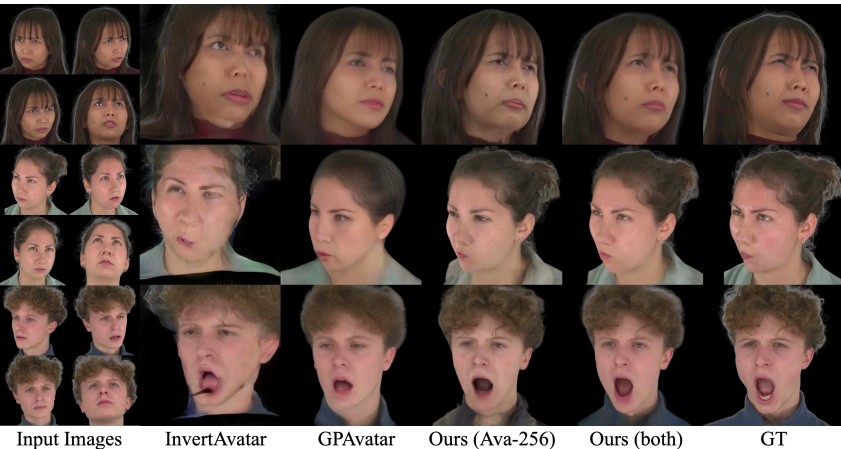

Input Images    InvertAvatar    GPAvatar    Ours (Ava-256)    Ours (both)    GT

Figure 4: Qualitative reconstruction comparison on Nersemble held-out subjects.

similarity (CSIM). Furthermore, we leverage Average Keypoint Distance (AKD) based on a facial landmark detector (Jin et al., 2021) to evaluate the reconstruction and motion accuracy.

## 4.2 MAIN RESULTS

**Quantitative Comparisons.** We quantitatively evaluate our FastGHA method on a reconstruction task using held-out data from both the Ava-256 and Nersemble datasets. Results are shown in Tab 1, comparing our method to the baselines across all metrics. Note that, for both InvertAvatar and GPAvatar, we use the official released code for comparison. For Avat3r, the code is not yet released but we thank the authors for providing their results on Ava-256 for us to compare to (unfortunately, their results on Nersemble data were not available).

The held-out data for the Ava-256 comparison includes 1000 different expressions from 12 subjects from different viewpoints. For Nersemble, we sampled 600 images from 4 subjects. We evaluate all methods at $512 \times 512$ pixels resolution, and for all results, we use four input images. For a fair comparison to Avat3r, which only used Ava-256 for training, we also train one version of our model on only the Ava-256 dataset (without Nersemble) and then show two separate results, one with only Ava-256 training data and one with both Ava-256 *and* Nersemble training data (labelled "both" in the table and figures) to compare against the other methods.

As shown in the table, our FastGHA method outperforms the baseline state-of-the-art approaches with a large gap in reconstruction quality (PSNR, SSIM, LPIPS) and identity preservation (CSIM), with equal or better performance on facial keypoint similarity (AKD). With both Ava-256 and

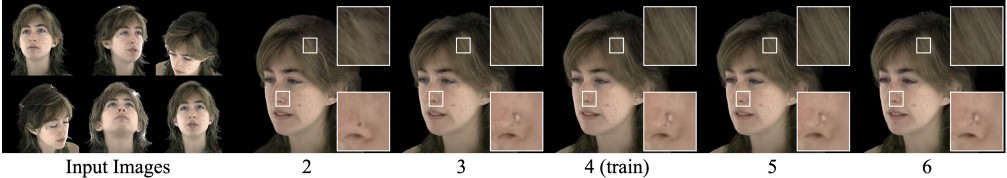

| Input Images | 2 | 3 | 4 (train) | 5 | 6 |

Figure 5: Analysis on the different number of input views. Two inputs means the first two in the top row, three inputs means the first three, four inputs adds the first image in the bottom row, and so on.

Table 2: Running time with different number of input images of our method. All results exclude the time for obtaining camera and expression parameters that can be calculated in advance.

| Number of Input | 2 | 3 | 4 | 5 | 6 |
|---|---|---|---|---|---|
| Reconstruction Time (s) | 0.68 | 0.83 | 0.98 | 1.15 | 1.40 |
| Animation Speed (FPS) | 128 | 83 | 62 | 48 | 32 |

Nersemble datasets in training, our method performs better than when training on Ava-256 alone, demonstrating that we can improve generalization by training on more data.

**Qualitative Comparisons.** We also show visual comparisons of held-out subject reconstructions using FastGHA and the baseline methods. Three subjects from the Ava-256 test set are shown in Figure 3, and three from Nersemble are shown in Figure 4. Note, again, that we do not have Avat3r results for the Nersemble test set.

This qualitative comparison shows that InvertAvatar and GPAvatar are unable to produce high fidelity results. Among the previous methods, only Avat3r can recover plausible photoreal avatars; however, it struggles with the identity recovery and does not match the target expression as well as our method. Important details like the hairstyle, skin texture and jewelry (earrings and nose rings) are better recovered by our FastGHA.

**Analysis on the Number of Input Views.** Although our method is trained and mostly evaluated with four input views, our architecture supports any number of input images. Figure 5 illustrates the reconstruction quality for a variety of input views, from two to six. Note how the overall reconstruction of the avatar is agnostic to the exact number of inputs, and only fine-scale details are missing when we have only sparser views. In particular, using the first two input images only, the subject's left-side of the face is unseen, and details are hypothesized by our method. Already by adding the third input view, left-side details are recovered.

We also measured the reconstruction time and real-time animation speed given the different number of inputs, as shown in Table 2. Because we infer per-pixel Gaussian attributes, the number of Gaussians and thus the reconstruction time both increase rather linearly with the number of input views, and the animation speed decreases accordingly, although still within real-time frame-rates even for six input images.

**Analysis of Input Expressions.** Since our model is trained with arbitrary input views and expressions, we analyze the impact of different input expressions on the reconstructed canonical Gaussian and animated Gaussian head in Figure 6. There are slight differences in the animated results caused by the variation in input expressions. Nevertheless, the model consistently produces accurate expressions and stable animations even when the four input views have significantly different expressions. Moreover, even though our method is trained end-to-end without explicit supervision on the canonical Gaussian, we observe that our model is able to disentangle identity and expression, by regressing the Gaussian avatar in a learned canonical representation. Specifically, when the four input images have different expressions, the model tends to learn a "mean" canonical Gaussian that averages the variations. While when all input views share similar expressions, the canonical Gaussian would preserve that expression to some extent but tends to be close to a neutral face.

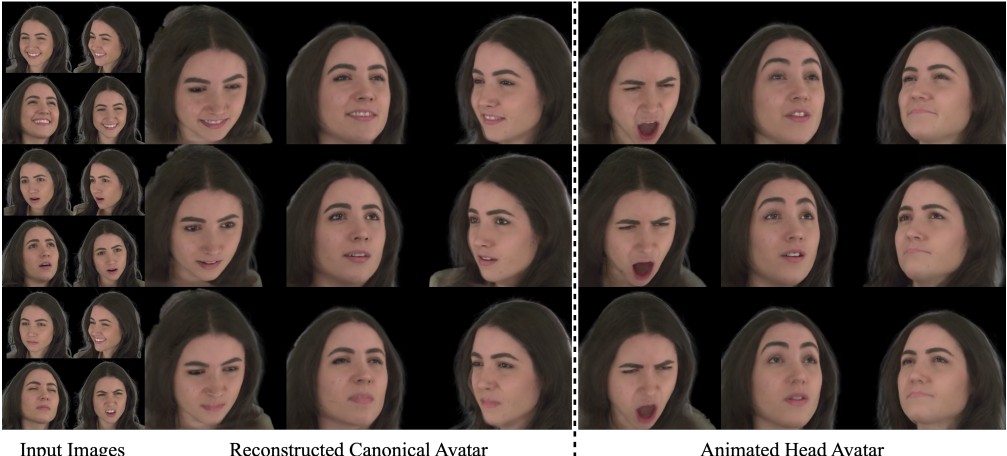

Input Images          Reconstructed Canonical Avatar          Animated Head Avatar

Figure 6: Visualization of the canonical and animated Gaussian head with different input expressions.

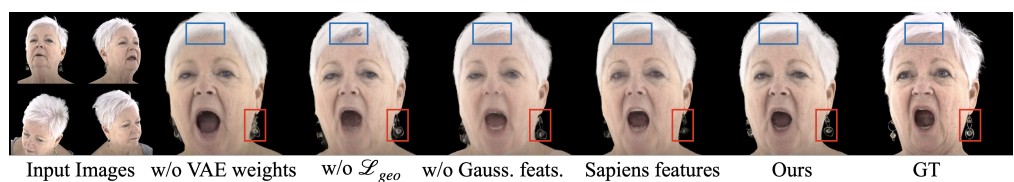

Input Images   w/o VAE weights   w/o $\mathcal{L}_{geo}$   w/o Gauss. feats.   Sapiens features   Ours   GT

Figure 7: Ablation study showing the importance of each of our design decisions.

## 4.3 ABLATION STUDY

We perform quantitative and qualitative ablations on several of our design decisions. The ablations are performed on the Ava-256 dataset. Note that the test set used for the ablation study differs from the one used in the main results, where the input images all share the same expressions as in Avat3r. The ablations are conducted on a separate test set where input images are with different expressions, which aligns with our training paradigm. Quantitative results are shown in Table 3, with qualitative visual comparisons given in Figure 7 and the supplemental video. Specifically, we evaluate the following alternative strategies for our network and training loss:

**w/o VAE weights.** In this version we do not use a *pre-trained* SD-Turbo VAE for our (frozen) encoder and (fine-tuned) decoder, but instead train the encoder and decoder from scratch using the same architecture. This experiment is inspired by Avat3r, who train their own encoder/decoder. Superior rendering quality is achieved by including the pre-trained weights as we do in our method.

**w/o $\mathcal{L}_{geo}$.** Here we show that without $\mathcal{L}_{geo}$ based on VGGT in our training loss function, we have worse 3D consistency and artifacts in the reconstruction.

**w/o per-Gaussian features.** Removing the learned per-Gaussian features from our network largely affects the animation accuracy (please refer to the supplemental video).

**Sapiens features.** We also trained a version of our network using Sapiens features (Khirodkar et al., 2024) in place of DINOv3, following Avat3r. Using the DINOv3 features provides better results.

## 4.4 APPLICATIONS

**Extreme Poses and Expressions.** We first showcase the capability of our method by animating and rendering the head avatar in extreme expressions and large view angles in Figure 8. In particular, we render novel views reaching 90°, even though such poses lie far outside the distribution of the input images.

Table 3: Ablation results on the Ava-256 dataset.

| Method | PSNR ↑ | SSIM ↑ | LPIPS ↓ | CSIM ↑ | AKD ↓ |
|---|---|---|---|---|---|
| w/o VAE weights | 20.789 | 0.738 | 0.251 | 0.681 | 5.487 |
| w/o $\mathcal{L}_{geo}$ | 21.132 | 0.741 | 0.239 | 0.687 | 5.049 |
| w/o per-Gaussian features | 21.053 | 0.740 | 0.245 | 0.690 | 5.216 |
| Sapiens features | 21.081 | 0.741 | 0.241 | 0.689 | 5.021 |
| Ours | **21.274** | **0.745** | **0.237** | **0.704** | **4.996** |

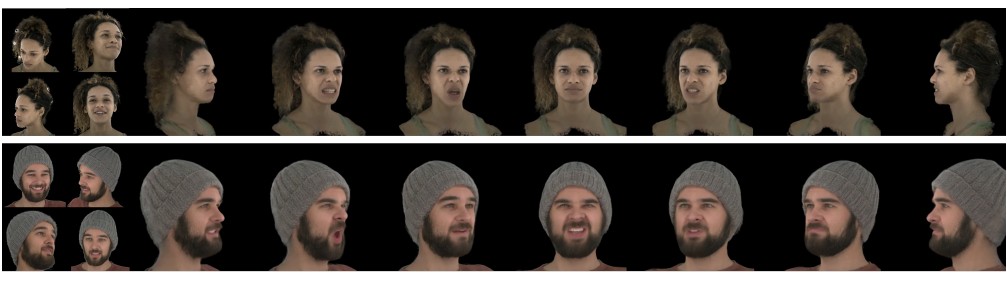

Input Images                                   Animated Head Avatar

Figure 8: Animation under large poses and expressions.

**In-the-wild Results.** We end demonstrating our method on the practical application of reconstructing 3D Gaussian avatars from candid photos in-the-wild. Different examples are shown in Figure 1. Our model supports portrait images captured from phone scans, monocular video frames, and images generated from a multi-view diffusion model. Even though the input images are outside the distribution of lighting and camera angles from the in-studio training datasets, our method has learned to generalize and creates realistic avatars that can be animated and rendered from novel views.

## 5 DISCUSSION

In this work we present FastGHA, a new feed-forward method to build a high-quality Gaussian avatar from a sparse number of input views. Our experiments have shown that our novel network design improves over both quality and efficiency compared to current state-of-the-art methods. In particular, FastGHA allows avatar reconstruction in under one second, followed by real-time animation and rendering, and can be applied to in-the-wild images.

**Limitations.** Our method shares similar limitations as Avat3r, in that we rely on accurate camera parameters as input, and our reconstructed avatars retain the same shading as the input images. Additionally, for our expression code parameterization we chose the FLAME expression space, as it offers intuitive artistic control over the resulting avatars. Unfortunately it does not allow to control the tongue motion. Other expression parameterizations like the one from Ava-256 could be used, but are less easily controllable for novel animation.

**Ethical Considerations.** Given only a small number of images, our method allows users to generate synthetic animations of a person. This technology should only be used with the consent of the subject. We do not support any unethical use of our method.

## 6 REPRODUCIBILITY STATEMENT

We strive to provide all the details required to reproduce our approach, including the architecture overview (Section 3.1), loss function details (Section 3.2), ablations on alternate variations (Section 4.3), and fine-level architecture details (in supplemental document).

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

# A  APPENDIX

In this supplementary material, we provide more details about the network architecture, data processing, and more information on our training details. We recommend watching the supplementary video for even more results.

## A.1  NETWORK ARCHITECTURE

We describe more implementation details on the network architecture used in Section 4.1 of the main paper. In Table 4, we present the details of our network architecture and the hyperparameters setting for our FastGHA. We freeze the DINOv3 and Diffusion VAE encoder for feature extraction and faster learning. For the VAE decoder, we extend the input layer dimension to 512 to match the output token dimension of the Multi-view Transformer. We also change the middle attention layers to cross-view attention for better view consistency. The VAE decoder is meant to regress pixel-wise Gaussian attributes, which includes position $\mathbf{X}$, multi-channel color $\mathbf{C}$, rotation $\mathbf{Q}$, scale $\mathbf{S}$, opacity $\alpha$, and per-Gaussian feature $\mathbf{f}$, in total 44 dimensions. Thus, we also expand the output layer dimension to predict all these attributes for each pixel. Following previous work Giebenhain et al. (2024); Dhamo et al. (2024), we leverage an MLP to predict Gaussian deformations from an expression code. To capture fine details of the motion, we apply sinusoidal positional encoding on the Gaussian position $\mathbf{X}$.

## A.2  TRAINING DETAILS

**Dataset processing.** We use large multi-view video dataset Ava-256 and Nersemble for training our model. In Ava-256 dataset, we leave 30 identities for testing, while in Nersemble dataset, we use 300 subjects for training. To factor the head pose inside the camera parameters, we use the head poses provided from the tracked mesh for the Ava-256 dataset. For the Nersemble dataset, we use a photometric head tracker based on VHAP (Qian et al., 2024) to fit FLAME parameters with multi-view images and known camera parameters. Then, we infer the head pose transformation from the FLAME parameters so that the network does not have to predict them.

**Input settings.** We train our network with four view images as input. Following Avat3r (Kirschstein et al., 2025), we leverage k-farthest viewpoint sampling during training for both Ava-256 and Nersemble dataset to ensure a reasonable distribution of these images. Moreover, we take input images from different timesteps to improve the robustness of the model. Specifically, we first sample a random camera and then use the farthest point sampling to collect the remaining 7 cameras for training and supervision. Then, for the input, we randomly select 4 timesteps inside the same sentence and one timestep for supervision.

**VGGT Geometry Loss.** To enhance the 3D geometry consistency and accelerate the training, we leverage the 3D prior from a large reconstruction model VGGT as supervision during training. VGGT is able to reconstruct a 3D scene from arbitrary number of inputs with predicted camera parameters, depth map, point map etc. However, we found out that the quality of the reconstructed camera and point cloud is not high enough and inconsistency exists. Thus, we take the 3d prior as a restrictive loss instead of an input for skip connection in Avat3r. Since VGGT is a large foundation model, we pre-compute the point maps for all camera views of each timestep. To get the point cloud that are aligned with the dataset camera, we use the 2d landmark along with the corresponding 3d keypoint in the tracked mesh. Specifically, we choose one frontal view and learn a transformation between the VGGT keypoint and tracked mesh keypoints for given facial landmarks. Then, we could transform and use the transformed point map as geometry supervision during training. Note that here we use the depth map to calculate the loss since we have the ground-truth camera parameters.

## A.3  ADDITIONAL RESULTS

### A.3.1  MORE ABLATION RESULTS

We analyze the contribution of DINOv3 feature map, and the efficacy of per-Gaussian feature dimension here. The quantitative results are listed in Table 5, and we show qualitative comparisons in Figure 9. We observe a noticeable degradation of shape consistency when the DINOv3 feature map

Table 4: Hyperparameters.

| | Hyperparameter | Value |
|---|---|---|
| | Input image resolution | $512 \times 512$ |
| Input & Output | Gaussian attribute map resolution | $512 \times 512$ |
| | Train render resolution | $512 \times 512$ |
| | DINOv3 version | dinov3-vitl16 |
| | DINOv3 feature size | $V \times 256 \times 64 \times 64$ |
| Feature Extraction | Diffusion VAE version | SD-Turbo |
| | VAE encoder feature size | $V \times 4 \times 64 \times 64$ |
| | Camera ray size | $V \times 6 \times 64 \times 64$ |
| | ViT patch size | $8 \times 8$ |
| | Hidden dimension | 512 |
| Multi-view Transformer | Self-attention layers | 8 |
| | Input token size | $266 \times V \times 64 \times 64$ |
| | Output token size | $512 \times V \times 64 \times 64$ |
| | Dimension of expression code | 120 |
| | Dimension of per-Gaussian feature | 32 |
| Deformation MLP $\mathcal{D}$ | Hidden dimension | 256 |
| | Number of hidden layers | 6 |
| | MLP activation | LeakyReLU |

Table 5: More ablation results on the Ava-256 dataset.

| Method | PSNR ↑ | SSIM ↑ | LPIPS ↓ | CSIM ↑ | AKD ↓ |
|---|---|---|---|---|---|
| w/o DINOv3 | 21.068 | 0.748 | 0.255 | 0.651 | 5.329 |
| per-Gaussian feature-16 | 21.220 | 0.739 | 0.251 | 0.672 | 5.190 |
| per-Gaussian feature-64 | **21.420** | **0.755** | 0.243 | 0.697 | **4.933** |
| Ours (DINOv3 + feature-32) | 21.274 | 0.745 | **0.237** | **0.704** | 4.996 |

is removed. Moreover, we analyze the performance with different choices of the per-Gaussian feature dimension. Results show that a higher dimension of per-Gaussian feature only brings marginal improvement, while decreasing the dimension noticeably degrades the animation accuracy.

### A.3.2   IN-THE-WILD RESULTS

Although our model is trained with laboratory datasets with ground-truth calibrated cameras, it generalizes well to in-the-wild input images with unknown camera parameters. In Figure 1, we show several use-cases of FastGHA, including portrait images captured from phone scans, monocular video frames, and images generated from a multi-view diffusion model (Taubner et al., 2025). Specifically, for images lacking camera calibration, we use an off-the-shelf monocular FLAME tracker (Zielonka et al., 2022; Giebenhain et al., 2025) to estimate the camera parameters with face meshes under the same canonical space. While the estimated cameras inevitably contain errors, our model still produces high-quality animations, indicating that our model is robust to moderate perturbation of cameras and generalizes well to unseen subjects, lighting conditions, and even cartoon characters.

### A.3.3   COMPARISON WITH ONE-SHOT METHODS

To evaluate the performance of our model with a single input image, we show comparisons with state-of-the-art 3D-aware one-shot methods, including GAGAvatar (Chu & Harada, 2024) and LAM (He et al., 2025). Both GAGAvatar and LAM rely on a 3D morphable model to reconstruct

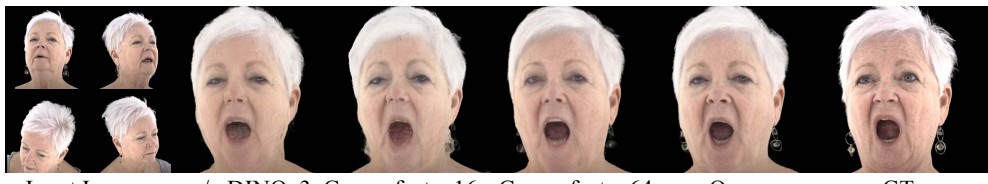

Input Images  w/o DINOv3  Gauss. feats.-16  Gauss. feats.-64  Ours  GT

Figure 9: More qualitative ablation of our design choices.

Table 6: Quantitative comparison with one-shot methods on the Nersemble dataset.

| Method | PSNR ↑ | SSIM ↑ | LPIPS ↓ | CSIM ↑ | AKD ↓ |
|---|---|---|---|---|---|
| GAGAvatar | 20.639 | 0.760 | 0.296 | 0.657 | 5.262 |
| LAM | 20.008 | 0.756 | 0.288 | 0.557 | 6.328 |
| Ours (one) | 19.299 | 0.729 | 0.327 | 0.497 | 8.249 |
| Ours (one+finetuned) | **23.002** | **0.790** | **0.263** | **0.730** | **4.725** |

and animate the Gaussian avatar from a single image. Moreover, GAGAvatar performs pixel-level post-processing with a 2D neural network, so it is not fully 3D. Quantitative and qualitative results are shown in Table 6 and Figure 10 respectively. Here, for each test subject in the Nersemble dataset, we select one frontal image as input. For GAGAvatar, we use their official FLAME tracker for data processing. For LAM, we use the same processed data as in our model.

Since our model is trained using four input images, directly applying it to a single input image results in holes in face regions, especially the unseen part. For a fair comparison, we fine-tune our model under the one-shot scenario for another 30k steps, which takes about 6 hours and denote it as "Ours (fine-tuned)". As can be seen in both the quantitative and qualitative results, our method achieves better identity preservation and expression accuracy, and the fine-tuned model can also hallucinate unseen areas in the single image input.

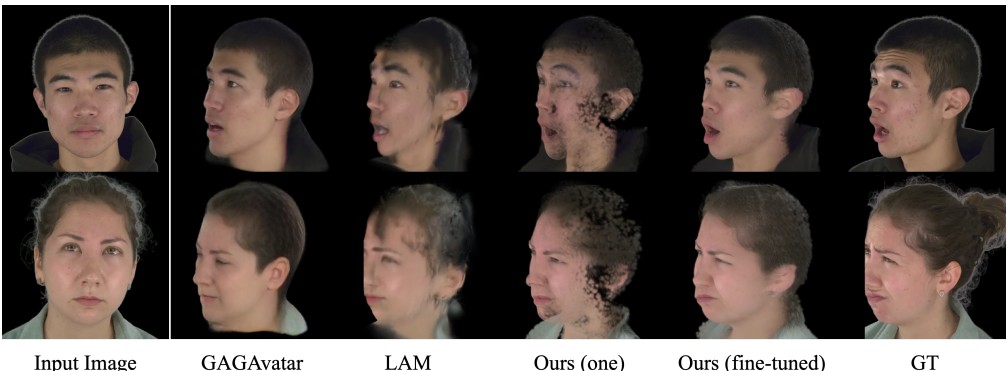

Input Image  GAGAvatar  LAM  Ours (one)  Ours (fine-tuned)  GT

Figure 10: Qualitative comparison with one-shot methods.

## A.4 Analysis of Model Design and Performance

### A.4.1 Analysis of Gaussian Offsets

As mentioned in the main paper, we predict the offsets of both per-Gaussian position and color from an MLP to model the facial dynamics. Here, we further evaluate the effectiveness of learning color offsets by comparing animation results with and without color offsets prediction. As can be seen in Figure 11, learning the color offsets is important for modeling fine-grained appearance, including wrinkles, skin details, and the teeth region when animating the avatar.

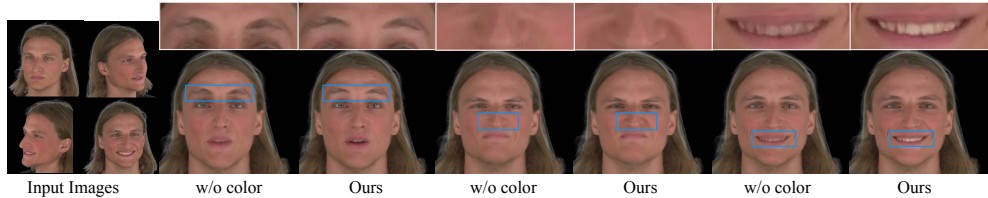

| Input Images | w/o color | Ours | w/o color | Ours | w/o color | Ours |

Figure 11: Effect of learning color offsets for each Gaussian.

### A.4.2 ANALYSIS OF NUMBER OF INPUT VIEWS

In Figure 12, we show how the performance of our model scales with the number of input views on both the Ava-256 and the Nersemble dataset. The results show a clear performance improvement as more input views are provided. However, more input views leads to higher computation cost and longer runtime inside the cross-attention network and the rendering of more number of Gaussian primitives. Moreover, the performance gain of using more than four input views becomes marginal.

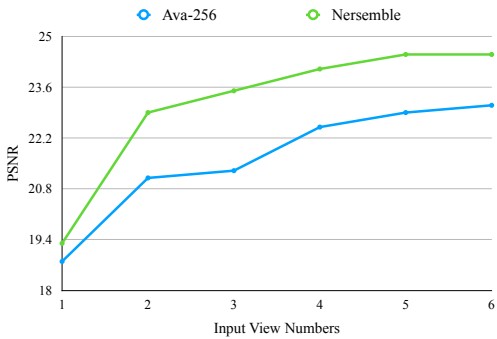

Figure 12: Achieved reconstruction quality with different number of input views, averaged over all subjects in the respective dataset.

### A.4.3 MORE ANALYSIS OF INPUT EXPRESSIONS

Here, we show more visualizations of the reconstructed canonical head and animated Gaussian head in Figure 13. The canonical Gaussian head is fully learned by the model without supervision, and a deformation MLP is trained to predict offsets relative to this space. We identify two factors that make the network converge to a relatively neutral canonical Gaussian head even when the input images contain expressions. First is the dataset distribution. Most of training frames depict neutral or near-neutral expressions. As a result, the model is encouraged to predict a canonical Gaussian representation that minimizes deformation inside the dataset. Second is our expression-driven deformation design. We use the FLAME expression paramater as driving signal for the deformation network to predict Gaussian offsets. Under such circumstances, a neutral-like canonical shape could provide a more stable reference for animation driven by the FLAME expression parameters.

### A.4.4 ANALYSIS ON CAMERA DEGRADATION

Here we evaluate the robustness of our model to camera perturbations by gradually adding noise to the ground-truth camera poses. As shown in Figure 14, inaccurate camera inputs noticeably degrade the multi-view consistency during animation. However, our model remains stable for small levels of camera perturbations, which is further supported by the in-the-wild examples in **??** where camera parameters are obtained through monocular optimization. Since our model is trained on datasets with accurate calibrated cameras, one potential direction to improve the robustness of the model is to add camera noise augmentation during training or alternatively developing an architecture that requires no explicit dependence on camera parameters. We leave this for future work.

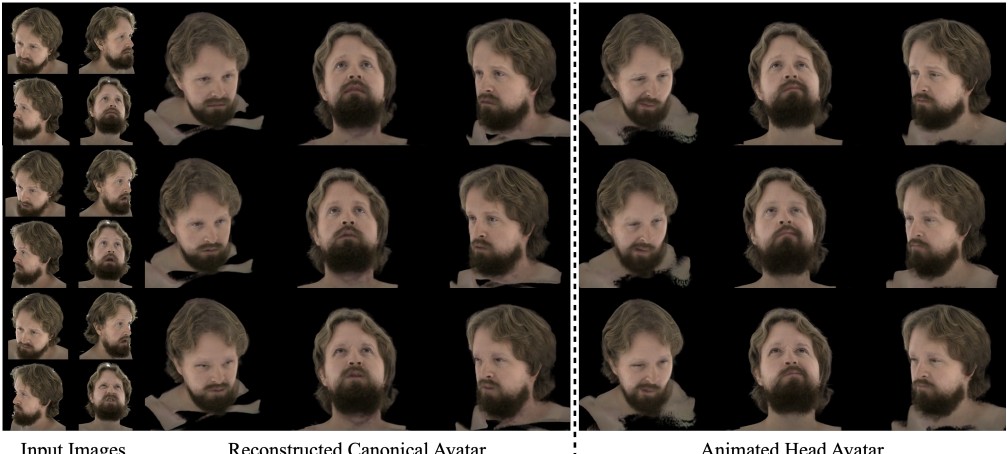

Input Images     Reconstructed Canonical Avatar     Animated Head Avatar

Figure 13: More visualization of the canonical and animated Gaussian head with different input expressions.

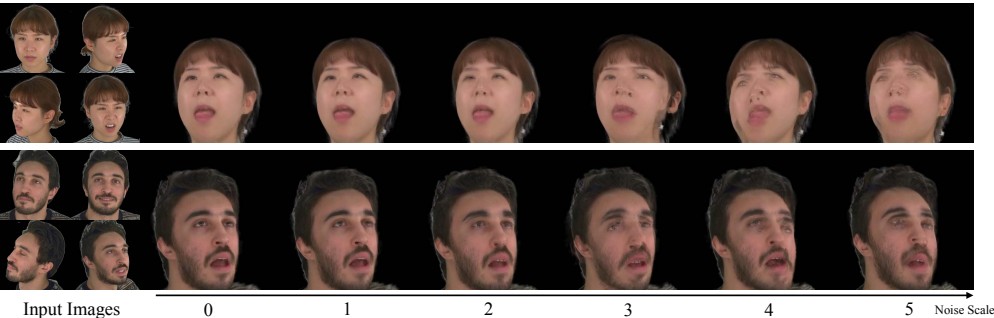

Input Images   0   1   2   3   4   5   Noise Scale

Figure 14: Impact of noise in the input camera poses on the quality of the generated avatar.

