# OpenReview forum: "FastGHA: Generalized Few-Shot 3D Gaussian Head Avatars with Real-Time Animation"
_ICLR.cc/2026/Conference — ICLR 2026 Poster_

### Official Review · Reviewer_4bsZ · 2025-10-29

**Soundness:** 3
**Presentation:** 3
**Contribution:** 3
**Rating:** 4
**Confidence:** 5

**Summary:**

This paper introduces a feed-forward framework for reconstructing animatable 3D Gaussian head avatars from only a few input images. The model first reconstructs a canonical Gaussian head using multi-view features extracted from pre-trained DINOv3 and Stable Diffusion VAE encoders, fused by a transformer-based module. Then, a lightweight MLP conditioned on FLAME expression codes predicts per-Gaussian deformations for real-time facial animation. Additionally, a geometry prior from the VGGT is employed as regularization to improve 3D consistency.

**Strengths:**

- Clear two-stage design that separates canonical reconstruction and dynamic animation.

- Efficient inference: <1s reconstruction and >60 FPS animation.

- Use of VGGT geometry prior enhances structural consistency.

- Outperforms Avat3r quantitatively in controlled datasets.

- Technically elegant combination of DINOv3, diffusion VAE, and Gaussian Splatting.

**Weaknesses:**

-  Methodological innovation over Avat3r is relatively incremental, primarily combining known modules (DINOv3, VAE, VGGT) rather than introducing a fundamentally new approach.
- Strong dependency on accurate camera poses, which significantly limits real-world usability. The method cannot operate from uncalibrated or monocular video, unlike many recent “in-the-wild” approaches, even though the paper acknowledges it in the limitations.

- In-the-wild results are weak and underexplored. Only a few examples are shown, and they exhibit degraded appearance and noticeable artifacts. The paper should provide more analysis on why generalization drops and how to mitigate it.

- Mouth region artifacts are frequent (see Figs. 2–3 and video results), with incomplete geometry and texture distortions, suggesting the method struggles with open-mouth or speaking expressions.

- Most visualizations show static reconstructions, not novel-view or novel-expression results, which are more important in avatar applications.

- The paper lacks convincing results for large-viewpoint novel renderings; side-view artifacts are visible in the supplementary video, contradicting the claim of a fully 3D avatar.

**Questions:**

- How does the model behave if camera calibration is slightly inaccurate or unavailable? Could the method be extended to work with monocular or unposed multi-view inputs?

- Can diffusion-based priors help improve in-the-wild generalization?

- Please provide more quantitative metrics for novel-view (side/back-view) and novel-expression performance.

---

> ### Author Response · Authors · 2025-11-26
> **Author Response to Reviewer 4bsZ**
>
> We greatly appreciate your review including the identified strengths of our work and your constructive comments for improvement.  We hope our response answers your questions, and addresses your concerns.
>
> > W1: Methodological innovation over Avat3r is relatively incremental
>
> We understand your perspective that our approach is indeed inspired by the architecture of Avat3r, but we feel that the improvements we propose do provide important technical advantages (e.g. in animation speed and quality).  As such, we believe other researchers will find our work insightful and will hopefully build upon it further.
>
>
> > W2&Q1: Strong dependency on accurate camera poses, which significantly limits real-world usability. How does the model behave if camera calibration is slightly inaccurate or unavailable? Could the method be extended to work with monocular or unposed multi-view inputs?
>
> Although our model is trained with laboratory datasets with ground-truth calibrated cameras, it generalizes well to in-the-wild input images where camera parameters are unavailable as shown in Figure 7, the new Figure 10 and the supplementary video.  Specifically, for images lacking camera calibration, we use an off-the-shelf monoculor FLAME tracker to estimate the camera parameters with face meshes under the same canonical space. While the estimated cameras inevitably contain errors, our model still produces high-quality animations, indicating that our model is robust to moderate perturbation of cameras and has strong generalization to unseen subjects. We further evaluate the robustness of our model to camera perturbations by gradually adding noise into the ground-truth camera poses. We present the visualization in Figure 16 and the supplementary video. Results show that an inaccurate camera leads to inconsistency during animation, but the model remains stable under small perturbations, which is consistent with the in-the-wild results.
>
> > W3: More in-the-wild results and provide more analysis on why generalization drops and how to mitigate it.
>
> We present additional in-the-wild results including phone-scan portraits, monocular video frames, and images generated by a multi-view diffusion model in Figure 10 and the supplementary video.  Results demonstrate that our method has strong generalization across subjects, lighting, and even stylized inputs. However the performance is sometimes lower than the test set in Ava-256 and Nersemble. One reason is the moderate camera pose errors introduced by monocular camera estimation, as discussed earlier. A promising mitigation strategy is to incorporate camera-noise augmentation during training or to explore architectures that eliminate explicit camera dependence. Moreover, our method is trained solely on laboratory datasets with limited identity numbers, thus involving more training data from large-scale or synthetic datasets could further improve generalization.
>
> > W5&W6: More visualization for novel-view, novel expression and large-viewpoint renderings. Most visualizations show static reconstructions, not novel-view or novel-expression results, which are more important in avatar applications.
>
> We provide more visualizations of large-view and novel-expression synthesis in the added Figure 9 and in the supplementary video. In particular, we render head avatars under extreme expressions and at view angles reaching 90 degrees, which lie far outside the distribution of the input images. Moreover, all subjects, animations, and camera trajectories in the video are unseen during training, thereby demonstrating novel-view, novel-expression, and large-viewpoint generalization beyond the training distribution.
>
> > Q2: Can diffusion-based priors help improve in-the-wild generalization?
>
> Yes, diffusion-based priors can potentially improve in-the-wild generalization. Recently, diffusion models have demonstrated strong generalization to images with various pose, lighting, and identities. One direction is to leverage a pretrained multi-view diffusion model to synthesize consistent input views during inference, and provide more cues for regions not observed in the input. Another option is to incorporate diffusion-based refinement, such as the DifFix+ work as a post-processing module to correct artifacts in Gaussian renderings.
>
> > Q3: Please provide more quantitative metrics for novel-view (side/back-view) and novel-expression performance.
>
> The quantitative metrics in Table 1 already evaluate novel-view and novel-expression performance. The comparisons are conducted on the test set of each dataset, where all the subjects, camera poses and expressions are unseen during training. Therefore, the PSNR/SSIM/LPIPS/CSIM/AKD scores have demonstrated the model’s ability to generalize to novel viewpoints and novel expressions.

---

> > ### Comment · Reviewer_4bsZ · 2025-11-28
> >
> > After reading the author's response and the newly added results, I appreciate the detailed clarifications and additional experiments. Most of my major concerns, especially regarding camera robustness, in-the-wild generalization, and novel-view rendering, have been addressed. The expanded visualizations and analyses strengthen the paper and improve my confidence in the method’s reliability.  A minor concern remains regarding the lack of fine dynamic wrinkles, though this is understandable given the model design and does not affect my overall assessment.
> > I am therefore raising my score accordingly.
> >
> > After reading other reviewers’ comments, I do have one new question: since the VGGT pseudo-GT is precomputed offline and reliable camera parameters are available, why not use a heavier and more accurate MVS reconstruction method instead of VGGT as the geometric prior? Clarifying this design choice would further strengthen the paper.

---

> > > ### Author Response · Authors · 2025-11-28
> > > **Thanks for the Reviewer's efforts**
> > >
> > > We are very glad that our response has addressed most of your concerns, and we sincerely appreciate your positive feedback and the constructive suggestions you provided, which help us further improve the paper. We are also grateful for your decision to increase the score.
> > > Regarding your question, we choose to use VGGT instead of a heavier MVS reconstruction method like Colmap because we take the geometric prior primarily as a restrictive guidance to accelerate and stabilize training, rather than as a high-accuracy supervision target. That's also why we assign a relatively small weight to the geometry loss. Compared with traditional MVS pipelines, VGGT is a fast, feed-forward reconstruction method without iterative optimization, while still providing sufficient and reliable geometry for human faces. This balance between efficiency and quality fits well with our training paradigm and enables us to scale to larger datasets and conduct rapid experiments.

---

### Official Review · Reviewer_XoYB · 2025-11-01

**Soundness:** 3
**Presentation:** 4
**Contribution:** 3
**Rating:** 8
**Confidence:** 4

**Summary:**

The paper introduces FastGHA, a novel feed-forward method designed to overcome the challenges of efficiently generating high-fidelity 3D Gaussian-based head avatars by utilizing only a few input images.

This approach notably supports real-time animation and achieves high fidelity through a two-stage pipeline: first reconstructing a canonical Gaussian head representation, which disentangles identity from the input expressions, followed by a deformation process. The system learns a per-pixel Gaussian representation, aggregating multi-view information using a transformer-based encoder that fuses semantic features from DINOv3 and color features from a Stable Diffusion VAE. For animation, the explicit Gaussian representation is augmented with learnable per-Gaussian features, and a lightweight MLP-based dynamic network is introduced to predict fine-scale 3D Gaussian deformations using descriptive expression codes, such as FLAME parameters. To improve the geometric consistency and robustness of the resulting 3D head, the method employs point maps derived from a pre-trained large reconstruction model (VGGT) as geometry supervision in the form of a regularization loss during training.

Extensive experiments confirm that FastGHA significantly outperforms existing state-of-the-art methods in both rendering quality and inference efficiency, enabling avatar reconstruction in less than one second.

**Strengths:**

- The paper presents a clear methodology and is well-structured.
- FastGHA employs a feed-forward approach to directly predict a per-pixel Gaussian representation. This design is explicitly chosen to enable instantaneous reconstruction of unseen subjects, avoiding lengthy per-identity optimization or template Gaussians rigged to 3DMMs required by many prior methods. Previous approaches struggled to handle fluffy elements like hair due to the rigging constraint.
- The method significantly addresses efficiency drawbacks by supporting real-time dynamic avatar animation. This speed is achieved through a lightweight MLP-based dynamic network that predicts fine-scale Gaussian deformations from expression codes, contrasting with computationally slower methods like those using cross-attention blocks. For a standard four-image input, the avatar reconstruction takes less than one second, and the animation speed remains high (e.g., 62 FPS for four inputs).
- FastGHA demonstrably outperforms existing state-of-the-art methods (including Avat3r, InvertAvatar, and GPAvatar) in reconstruction fidelity, showing a large gap in quantitative metrics such as PSNR, SSIM, LPIPS, and identity preservation.

**Weaknesses:**

- The paper identifies Avat3r as the "most related" state-of-the-art method. However, the crucial quantitative comparison table (Table 1) lacks the results for Avat3r on the Nersemble dataset. This prevents a full comparison, especially given that FastGHA's best performance is achieved when training on both Ava-256 and Nersemble ("Ours (both)").
- The animation pipeline depends on the accuracy of FLAME expression codes ($z_{exp}$) obtained using "off-the-shelf head tracking tools". Errors generated by these trackers when processing complex or extreme expressions (e.g., a wide open mouth or squinting, as shown in the input examples) could compromise the fidelity of the downstream deformation network $D$.

**Questions:**

- In the qualitative comparisons (as shown in video), I still observe flicker in the Gaussian primitives. Could the authors explain the cause of these artifacts? Would increasing the number of input views alleviate the issue? Including curves of performance versus number of views in the final version would help analyze the impact of few-shot inputs.

- The method reconstructs a canonical Gaussian Head even when the input images contain expressions, and the authors state that the canonical Gaussian is unsupervised. I am curious why this representation can be learned. Could the authors offer a deeper discussion? Is this related to using neutral faces during training, enabling the model to learn that, when the expression code is absent, it should output a canonical Gaussian?

- Could the authors discuss whether the pre-processing of VGGT—specifically the alignment of its reconstructed point cloud using 2D landmarks and corresponding 3D keypoints from a tracked mesh—introduces additional errors or complexities? In particular, how do inaccuracies in landmark detection and mesh tracking affect the reliability of the geometry prior and the regularization loss $L_{geo}$, given that the VGGT point cloud quality is reported as "not high enough" and "inconsistent" (Line 849-850)?

---

> ### Author Response · Authors · 2025-11-26
> **Author Response to Reviewer XoYB**
>
> We greatly appreciate your time, your positive review, and your constructive comments.  We hope that our response answers your questions and alleviates any concerns about our work.
>
> > W1: Lacks the results for Avat3r on the Nersemble dataset.
>
> Avat3r is not open-sourced, thus we couldn't run it on the Nersemble dataset ourselves. We contacted the authors before submission to request their evaluation outputs for both Ava-256 and Nersemble. However, they were only able to send their Ava-256 results (so far), and thus we report only those in the paper. We would be glad to include Nersemble results as soon as the code is available or if the authors can provide them.
>
> > Q1: Causes of flicker in the Gaussian primitives and would increasing the number of input views alleviate the issue? Including curves of performance versus number of views in the final version would help analyze the impact of few-shot inputs.
>
> We assume the flicker you refer to is either the subtle expression inconsistency during facial animation or the more noticeable flicker in the non-face area, such as the neck, shoulders? Please let us know if we misunderstood your observation. There are two main reasons for this flicker. One is the lack of temporal smoothness in our training. Our animation model is trained frame by frame, without any temporal constraints, thus the variations in the driving expression parameters can lead to subtle flicker in the animations occasionally.  Another reason is that the neck and the shoulder part is not modeled by the FLAME parameters, therefore these regions can have minor instability, especially under large pose changes. As shown in the supplementary video with 2-6 input views, the animation quality and the motion of the neck and shoulder is not affected by the input view numbers. One promising solution is to incorporate temporal smoothness losses during training, or to explicitly stabilize the non-face region.  We are happy to mention these improvements as future work.
>
> Regarding the number of input views, we have added a curve in Figure 13 to show how the performance of our model scales with the number of input views on both the Ava-256 and the Nersemble dataset. The results show a clear performance improvement as more input views are provided. However, more input views leads to higher computation cost and longer runtime inside the cross-attention network and the rendering of more number of Gaussian primitives. Moreover, the performance gain of using more than four input views becomes marginal. For these reasons, we adopt four input views for training.
>
> > Q2: Could the authors offer a deeper discussion on the canonical Gaussian head?
>
> The canonical Gaussian head is fully learned by the model without supervision, and a deformation MLP is trained to predict offsets relative to this space. Here we identify two factors that make the network converge to a relatively neutral canonical Gaussian head even when the input images contain expressions. First is the dataset distribution. Most of the training frames depict neutral or near-neutral expressions. As a result, the model is encouraged to predict a canonical Gaussian representation that minimizes deformation inside the dataset. Second is our expression-driven deformation design. We use the FLAME expression parameter as a driving signal for the deformation network to predict Gaussian offsets. Under such circumstances, a neutral-like canonical shape could provide a more stable reference for animation driven by the FLAME expression parameters.
>
> > Q3: Could the authors discuss whether the pre-processing of VGGT—specifically the alignment of its reconstructed point cloud using 2D landmarks and corresponding 3D keypoints from a tracked mesh—introduces additional errors or complexities?
>
> The pre-processing of VGGT for our training data is straightforward and doesn't introduce a landmark detection error.  This is due to the high-quality tracked meshes provided in the Ava-256 dataset and obtained using a multi-view FLAME tracker with ground-truth camera calibration for the NeRSemble dataset. Moreover, we don't perform any 2D landmark detection for the alignment. Instead, 2D landmarks are obtained by projecting the 3D mesh vertices to the image plane using the calibrated camera parameters. This avoids the landmark detection inaccuracies. While VGGT reconstructions can be noisy or inconsistent across views, we use them only as a weak restrictive loss rather than an input for skip connection as in Avat3r. This keeps the preprocessing simple and ensures that VGGT noise does not adversely affect training.

---

### Official Review · Reviewer_uEfC · 2025-11-01

**Soundness:** 3
**Presentation:** 3
**Contribution:** 3
**Rating:** 8
**Confidence:** 4

**Summary:**

The paper addresses the task of reconstructing an animatable 3DGS-based avatar from 4 posed input images.

Following the recent success of LRM-style approached, like Avat3r,, the method employs a multi-view transformer which use features from a stable diffusion VAE, DINOv2 and plucker ray embeddings to predict per-pixel Gaussian attributes.

Importantly, the set of gaussian attributes is extended to contain per-Gaussian latent features, similar to GHA and NPGA, which are then used to condition an MLP which animates the Gaussian based on FLAME expression code conditoning. Compared to Avat3r, the MLP is more light weight than the cross-attention based architecture, which is relevant for real-time applications.

Adding supervision against aligned VGGT depth predictions helps to produce more 3D consistent avatars.

The method is thoroughly evaluated against comparable sota baselines, and additionally shows examples on real-world images.

**Strengths:**

- The task of accessable avatar creation from a few selfie images finds importance in many down-stream applications. Therefore, the improved visual quality and faster animation speed immediatley become more significant.
- Smart and simple usege of pretrained VAE weights for encoder/decoder, which is also ablated to be beneficial. The same can be said for the VGGT supervision.
- Overall the architecture seems to be slightly simplified comapred to Avat3r, which seems quite helful for future improvements that build upon the presented method.
- The paper is well-written and easy to follow.
- The method is thoroughly evaluated against to most relevant baselines, and on two different datasets, leaving little room for doubt about the evaluation. Furthermore, the SOTA baselines are significantly outperformed.
-

**Weaknesses:**

- The main weakness of the paper that I am still seeing is the limited novelty, since it mainly introduces some technical changes compares to Avat3r. However, the quality and animation speed are improved, and the paper is well evaluted. Therefore, we can be sure that the method solidly advances the field on such a highly relevant task. Therefore, I don't mind the limited novelty.
- Currently, the method is limited by FLAME expression codes. However, anything else would likely be out-of-scope for this project, and it would make animation from 2D videos arguably harder, bc. obtaining FLAME codes is so easy.
- Somewhow the number in the ablation study don't match the main results. Is there an explanation for this? Otherwise, this gives the impression that the ablations were sloppily exectued.

**Questions:**

- The text states that the 4 input views have different facial expressions during training. However, all results show identitcal expressions, or quite similar expressions. Does the model still perform well with very different expressions?
- Fig. 5: would be interesting to see how much the canonical point cloud changes with different inputs, e.g. the first example has the mouth slightly open in canonical space, which might be caused by the input images.
-  Is the VGGT superivsion competed offline? How good do the recoinstructions look? In my experience they can be rather blurry for faces and exhibit many stiching artifacts. Could this have been done with e.g. COLMAP as well?

---

> ### Author Response · Authors · 2025-11-26
> **Author Response to Reviewer uEfC**
>
> We greatly appreciate your positive review and constructive comments, and we hope our response answers all your questions and addresses any further concerns.
>
> > W1: limited novelty over Avat3r.
>
> We understand your perspective that our approach is indeed inspired by the architecture of Avat3r, but we feel that the improvements we propose do provide important technical advantages (e.g. in animation speed and quality), as you also indicate in your review.  As such, we believe other researchers will find our work insightful and will hopefully build upon it further.
>
> > W2: limited by FLAME expression codes.
>
> As you correctly indicated, FLAME parameters are easy to obtain from video and intuitive to control by artists, hence our choice of parameterization.  We'd be excited to hear about alternatives you have in mind, and would list alternate control modes as future work.
>
> > W3: Somehow the number in the ablation study don't match the main results.
>
> Good point, and we apologize for the lack of clarity here. The inconsistency arises because different test sets were used for the main comparison experiments versus the ablation study. In the main comparison on the Ava-256 dataset, we show comparisons with Avat3r, which is not open-sourced yet. Therefore we reached out to the authors and obtained their test set and results. For fair comparison, all methods including ours are evaluated on their Avat3r test set, in which all input views share the same expression. On the other hand, ablation study results are conducted on a separate test set where input images have different expressions, which aligns better with our training paradigm. This allows us to accurately evaluate each design component of our model under the training conditions.  We will certainly clarify this in the final paper.
>
> > Q1: Does the model still perform well with very different expressions?
>
> Yes, we present additional examples in Figure 14 and the supplementary video, where the input views either have the same input expressions or very different expressions to further analyze the impact of input expressions to the animation. Results show that our model could consistently produce accurate expressions and stable animations even when the four input views have significantly different expressions.
>
> > Q2: How much the canonical point cloud changes with different inputs?
>
> We visualize the reconstructed canonical Gaussian from different input expression settings in Figure 15 and the supplementary video. We observe that when the four input images have different expressions, the model tends to learn a "mean" canonical Gaussian that averages the variations. While when all input views share similar expressions, the canonical Gaussian would preserve that expression to some extent but tends to be closer to a neutral face. We hypothesize that the model tends to learn a neutral-like canonical shape because this provides a more stable reference for animation driven by the FLAME expression parameters.
>
> > Q3: Is the VGGT superivsion computed offline? How good do the recoinstructions look? Could this have been done with e.g. COLMAP as well?
>
> Yes, the VGGT supervision is computed offline since VGGT is a large reconstruction model and is too heavy to run during training. During the VGGT inference, we also observed that the reconstructed camera and point cloud can be noisy and inconsistent across views. This is the main reason we use the VGGT geometry as a restrictive loss rather than an input for skip connection as in Avat3r. We also assign a small weight for this geometry loss to prevent the model from being affected by errors in the VGGT geometry. While COLMAP could also be used to get a geometry prior, we choose VGGT because it is a fast, feed-forward method and produces reasonable quality reconstructions on human faces without iterative optimization.

---

### Official Review · Reviewer_2Epj · 2025-11-03

**Soundness:** 3
**Presentation:** 3
**Contribution:** 3
**Rating:** 4
**Confidence:** 4

**Summary:**

This paper proposes FastGHA, which efficiently generates high-quality, real-time animated 3D Gaussian head avatars with few-shot inputs. The method fuses multi-view features through a Transformer encoder and utilizes Gaussian features and a lightweight MLP network to achieve real-time animation based on facial expression encoding. Simultaneously, geometric supervision is introduced to improve the geometric smoothness of the 3D head.

**Strengths:**

* The method demonstrates satisfactory few-shot reconstruction quality and visualization effects.
* The method's reconstruction design is reasonable, effectively integrating information from different views, and can be well driven by new expressions.

**Weaknesses:**

* This method may struggle with "unreasonable" user input and the paper does not show these results. Furthermore, reasonable input is difficult to define. For example, if user input from a particular perspective is missing, the result is currently unknown.
* This method may struggle to handle an arbitrary number of input viewpoints. Because it relies on VxHxW self-attention, too many views exponentially increase the computational cost of this part.
* Due to the linear increase in the number of Gaussians and the presence of the driving MLP, the driving speed decreases linearly with increasing viewpoint. Merging and controlling the total number of reconstructed Gaussians might be a good solution.

**Questions:**

* Why does the MLP need to modify the color of the Gaussians during animation? What would happen if it didn’t?
* The paper conducts ablation experiments on removing the VAE feature. What would be the result if only the VAE feature is kept and the DINO v3/Sapiens feature is removed?
* Removing the per-Gaussian feature shows a significant difference in results. What would happen with other dimension choice of the per-Gaussian feature, for example, 16 or 64?
* This method seems to have a hidden limitation: the requirement for a reasonable distribution of input viewpoints. Does this mean that the method lacks or has limited reconstruction ability when certain viewpoints are not covered? Structurally, the method can reduce the input to just one image. What is the performance when only a single frontal image is provided, or what if one frontal image and one side image, with the other side / up side missing?
* As a few-shot method, it is also reasonable and beneficial to discuss and compare with some of the latest one-shot methods, which can highlight the unique advantages of few-shot approaches. The authors could discuss and compare with methods such as GAGAvatar and LAM.
* Since the method is completely unsupervised in the canonical space, the statement “By design, the canonical head
is devoid of expression and thus does not correspond to any of the input expressions, allowing animation to happen downstream. ” is too arbitrary. It is possible for the model to learn a canonical space with an open mouth, and in fact, a canonical space with an open mouth may make it easier to model teeth than the closed-mouth shown in the Figure 1. What causes the model converges to a closed-mouth canonical space?
* There is no explanation for C_d and C_e in Equation 3. After reading, it seems refer to the DINO feature and encoder feature, but this should be explicitly stated.
* The additional Gaussian features used in animation MLP should not be claimed as "learnable", because it is not parameters of the model, but rather the result of inference. They are essentially the same thing as other inferred Gaussian parameters.

---

> ### Author Response · Authors · 2025-11-26
> **Author Response to Reviewer 2Epj (Part1)**
>
> We are grateful for the time you spent to provide your review and we hope our response fully answers your questions and alleviates any concerns about weaknesses.
>
> > Q1: Why does the MLP need to modify the color of the Gaussians during animation?
>
> We evaluated the role of color modification by comparing animation results with and without color desplacements prediction in Section A.4.1 and supplementary video. Results show that learning the color offsets is important for modeling fine-grained appearance, including wrinkles, skin texture and teeth region during animation. Without color offsets, the rendered face appears flatter and less expressive.
>
> > Q2&Q3: What would be the result if only the VAE feature is kept and the DINO v3/Sapiens feature is removed? What would happen with other dimension choices of the per-Gaussian feature, for example, 16 or 64?
>
> Thank you for raising this point. Below, we provided the ablation results for DINOv3 feature and using different dimensions for per-Gaussian features on the Ava-256 dataset. More qualitative results can be found in Section A3.1 and supplementary video. It can be observed that there are obvious degradation of shape consistency when DINOv3 feature is removed. On the other hand, results show that a higher dimension of per-Gaussian feature only brings marginal improvement, while decreasing the dimension noticeably affects the animation accuracy.
>
> | Method                  |    PSNR    |   SSIM    |   LPIPS   |   CSIM    |    AKD    |
> | :---------------------- | :--------: | :-------: | :-------: | :-------: | :-------: |
> | w/o DINOv3              |   21.068   |   0.748   |   0.255   |   0.651   |   5.329   |
> | per-Gaussian feature-16 |   21.220   |   0.739   |   0.251   |   0.672   |   5.190   |
> | per-Gaussian feature-64 | **21.420** | **0.755** |   0.243   |   0.697   | **4.933** |
> | Ours                    |   21.274   |   0.745   | **0.237** | **0.704** |   4.996   |
>
> > W1&Q4&Q5: Does the method lack or have limited reconstruction ability when certain viewpoints are not covered? What is the performance when only a single frontal image is provided, or what if one frontal image and one side image, with the other side / up side missing? it is also reasonable and beneficial to discuss and compare with some of the latest one-shot methods.
>
> We apologize for the insufficient clarity in the main paper. Our method has the ability to reconstruct and animate unseen areas in the input images. For example, in Figure 4 and the corresponding supplementary video we present the animation with only 2 input images, one frontal image and one right side image. Results show that our model successfully reconstructs the unseen side and the top of the head, although the quality of the hair on the unseen side is less accurate.
>
> We further showed results from only a single frontal image and added comparisons with state-of-the-art 3D-aware one-shot methods, including GAGAvatar and LAM in Section A.3.4.  Since our model is trained using four input images, directly applying it to a single input image results in holes in some regions. For a fair comparison, we fine-tune our model under the one-shot scenario for another 30k steps, which takes about 6 hours and denote it as "Ours(fine-tune)". Then for each test subject in the Nersemble dataset, we select one frontal video image as input. We put the quantitative results below and more qualitative comparisons can be found in the paper. As can be seen in both the quantitative and qualitative results, our method achieves better identity preservation and expression accuracy, and the fine-tuned model can also hallucinate unseen areas in the single image input.
>
> | Method              |    PSNR    |   SSIM    |   LPIPS   |   CSIM    |    AKD    |
> | :------------------ | :--------: | :-------: | :-------: | :-------: | :-------: |
> | GAGAvatar           |   20.639   |   0.760   |   0.296   |   0.657   |   5.262   |
> | LAM                 |   20.008   |   0.756   |   0.288   |   0.557   |   6.328   |
> | Ours (one)          |   19.299   |   0.729   |   0.327   |   0.497   |   8.249   |
> | Ours (one+finetune) | **23.002** | **0.790** | **0.263** | **0.730** | **4.725** |
>
> > Q6: What causes the model converge to a closed-mouth canonical space?
>
> During training, the canonical gaussian is fully unsupervised and the deformation MLP is designed to learn the offsets relative to this space. Here we conclude two factors contributing to the tendency of a closed-mouth canonical space. First is the dataset distribution. Most of the training frames are with neutral or lightly open mouths, biasing the learned canonical Gaussian head toward a neutral expression. Second is our expression-driven deformation design. We use the FLAME expression parameter as a driving signal for the deformation network. Thus a neutral-like canonical shape could provide a more stable reference for animation driven by the FLAME expression parameters.

---

> ### Author Response · Authors · 2025-11-26
> **Author Response to Reviewer 2Epj (Part2)**
>
> > W2&W3: This method may struggle to handle an arbitrary number of input viewpoints (computation cost), and the driving speed decreases linearly with increasing viewpoint.
>
> Your point is correct, with an increasing number of viewpoints our method would suffer from larger computational cost, more Gaussians and thus decreased speed.  That said, we have shown in Figure 4 and the new Figure 13 that only 4 or 5 well-distributed viewpoints are sufficient to obtain a realistic avatar, and beyond that (even with 6) the improvements in quality and coverage are negligible.  Hence, our method does not require (and we would not even recommend using) a large number of input views.
>
> > Q7: There is no explanation for C_d and C_e in Equation 3. The additional Gaussian features used in animation MLP should not be claimed as "learnable".
>
> We apologize for the lack of clarity and misleading statements in our paper. We have added the explanation and corrected the statement in the revised version.

---

> ### Comment · Reviewer_2Epj · 2025-11-27
>
> Thank you to the author for the clarification and additional experiments.
> I think they resolved many of my concerns, and the supplementary experiments also provided new insights.
> I'm also pleased to see that the authors have integrated the experimental results into the revised version.
> I will increase my score.

---

> > ### Author Response · Authors · 2025-11-28
> > **Thanks for the Reviewer's efforts**
> >
> > Thank you so much for acknowledging the updated results in the revised version. We sincerely appreciate your constructive feedback and are glad that the clarifications and supplementary experiments resolved your concerns. Your insights have been valuable in improving the paper, and we are grateful for your decision to increase the score.

---

### Author Response · Authors · 2025-11-26
**General Response to All Reviewers**

We sincerely thank all reviewers for their positive comments and valuable feedback on our paper, which is very encouraging for us. Following your suggestions, we have added new experiments, analysis, and explanations in both the revised paper (new text is marked in blue, and supplemental Sections A.3 and A.4 are brand new) and the supplementary video (addition.mp4). Below we summarize the updates:

### Additional Experimental Results

- We conducted more ablation studies on the DINOv3 features and the design of per-Gaussian feature dimension (A.3.1);
- We provided more results for in-the-wild images without camera calibration and animation with extreme poses and expressions to support the effectiveness of our method (A.3.2, A.3.3);
- We included new comparisons with the latest 3D-aware one-shot method,s including GAGAvatar and LAM (A.3.4).

### Analysis of Model Design and Performance

- We evaluated the importance of learning color offsets in the MLP, showing that it's vital for capturing fine-grained facial appearance (A.4.1).
- We added an analysis curve demonstrating how performance scales with the number of input views (A.4.2);
- We examined the effect of different input expressions on both the reconstructed canonical Gaussians and the animated head avatar, demonstrating the generalization ability of our model to diverse expressions (A.4.3);
- We analyzed the impact of camera perturbations and demonstrated that our model remains robust under moderate camera noise, consistent with results on in-the-wild images (A.4.4).

We also revised the paper to clarify ambiguous sentences, adding explanations to make the paper easier to understand.

Once again, we are grateful for all the constructive suggestions from the reviewers. We believe all these additions would significantly enhance the completeness and impact of our paper. We hope our additional individual responses address your concerns and we would be glad to receive any further feedback.

---

### Meta-Review · Area_Chair_gu8w · 2025-12-09

**Summary:**

The paper proposes FastGHA for efficient, few-shot 3D Gaussian head avatar creation. The initial reviews were mixed, with positive assessments of the efficiency/quality (uEfC, XoYB) balanced by concerns regarding incremental novelty (wrt Avat3r), robustness to uncalibrated cameras, and in-the-wild performance (2Epj, 4bsZ). The rebuttal works well: the authors provided new experiments on camera noise robustness, expanded in-the-wild visualizations, and clarified architectural choices. Consequently, two reviewers (2Epj and 4bsZ) expressed the willingness to raise their scores. The AC agrees that while the architectural novelty is moderate, the performance gain and practical utility warrant acceptance.

**Reviewer Concerns:**

## Addressed:
- Concern about the dependency on perfect calibration was addressed via a noise perturbation study and demonstrations on uncalibrated in-the-wild video. (Reviewer 4bsZ)

- Comparison with GAG and LAM were included and still showed advantage. (Reviewer 2Epj)

- Concerns about performance on unseen datasets and single/two-view inputs were addressed. (Reviewer 2Epj)

- Discrepancies in ablation numbers were explained as a difference in test set usage. (Reviewer uEfC)

## Outstanding issues:

- Reviewers uEfC and 4bsZ noted the similarity to Avat3r. This remains true, but they agreed the performance improvements justify acceptance.

- Minor flickering and lack of deep dynamic wrinkles remain a limitation of the method, acknowledged as future work.

**Reviewer Scores:**

- Reviewer 2Epj 4 ==> 6 (increase). Concerns were resolved.
- Reviewer uEfC 8 ==> 8 (unchage). Positive about the paper. Issues were minor/solved.
- Reviewer XoYB 8 ==> Score: 8 (unchage). Positive about the paper. Issues were minor/solved.
- Reviewer 4bsZ 4 ==> Score: 6 (increase). Concerns were resolved.

So most likely, all reviewers are towards accepting this paper after rebuttal.

---

### Decision · Program_Chairs · 2026-01-26

Accept (Poster)